# High Endothelial Venule with Concomitant High CD8+ Tumor-Infiltrating Lymphocytes Is Associated with a Favorable Prognosis in Resected Gastric Cancer

**DOI:** 10.3390/jcm9082628

**Published:** 2020-08-13

**Authors:** Soon Auck Hong, Hye Won Hwang, Min Kyoon Kim, Tae Jin Lee, Kwangil Yim, Hye Sung Won, Der Sheng Sun, Eun Young Kim, Yoon Ho Ko

**Affiliations:** 1Department of Pathology, College of Medicine, Chung-Ang University, Dongjak-gu, Seoul 06974, Korea; hsu108@cau.ac.kr (S.A.H.); hwhwang@caumc.or.kr (H.W.H.); taejilee@cau.ac.kr (T.J.L.); 2Department of Surgery, Chung-Ang University Hospital, College of Medicine, Chung-Ang University, Dongjak-gu, Seoul 06974, Korea; likeicetea@hanmail.net; 3Department of Hospital Pathology, Uijeongbu St. Mary’s Hospital, College of Medicine, The Catholic University of Korea, Seocho-gu, Seoul 06591, Korea; kangse_manse@catholic.ac.kr; 4Division of Medical Oncology, Department of Internal Medicine, College of Medicine, The Catholic University of Korea, Seocho-gu, Seoul 06591, Korea; woncomet@catholic.ac.kr (H.S.W.); ds-sun@daum.net (D.S.S.); 5Department of Surgery, Uijeongbu St. Mary’s Hospital, College of Medicine, The Catholic University of Korea, Seocho-gu, Seoul 06591, Korea; 6Cancer Research Institute, College of Medicine, The Catholic University of Korea, Seocho-gu, Seoul 06591, Korea

**Keywords:** gastric cancer, high endothelial venule, CD8+ tumor-infiltrating lymphocytes, prognosis

## Abstract

CD8+ tumor-infiltrating lymphocytes (TILs) play a major role in antitumor immunity. High endothelial venules (HEVs) are related to diverse immune cells in solid tumors. We analyzed CD8+ and Foxp3+ TILs in combination with HEVs to determine their prognostic role in advanced gastric cancer (AGC). We enrolled 157 patients with AGC in this study. The densities of CD8+ TILs and Foxp3+ TILs were calculated using immunohistochemical staining. HEVs were evaluated by MECA-79 expression. HEVs were identified in 60 (38.2%) cases and was significantly associated with an increased number of CD8+ TILs (*p* = 0.027) but not of Foxp3+ TILs (*p* = 0.455) and CD20+ TILs (*p* = 0.163). A high CD8+/HEV+ level was significantly associated with nodal metastasis (*p* = 0.048). In survival analysis, patients with high CD8+/HEV+ levels demonstrated the longest overall survival (OS) (*p* = 0.015). Furthermore, a high CD8+/HEV+ level was an independent prognostic factor in AGC (*p* = 0.011; hazard ratio (HR) = 0.435; 95% confidence interval (CI) = 0.245–0.837). HEVs were found to play an important role in antitumor immunity associated with CD8+ TILs in AGC. This analysis of HEVs and CD8+ TILs helps stratify patients with AGC and sheds light on tumor immunity.

## 1. Introduction

Gastric cancer (GC) is the sixth most common cancer and the second leading cause of cancer mortality worldwide [1]. Advanced gastric cancer (AGC) is one of the most aggressive cancers, with a five-year survival rate of less than 10% [2]. Immunotherapy has been the mainstay of treatment for various advanced cancers [3]. A recent study indicated that nivolumab, as a third-line setting in refractory GC, significantly prolonged survival compared to a placebo [4]. Immunotherapy alone or in combination with targeted therapy is emerging as a promising treatment for AGC.

Currently, immunotherapy focuses on restoring the antitumor immunity by effector T cells, including blocking immune checkpoints [5,6]. However, recent studies have focused on controlling antitumor immunity through the tumor microenvironment. High endothelial venules (HEVs) are the new biomarkers of tumor immunity [7,8]. HEVs are post-capillary venules characterized by active lymphocyte trafficking; they are usually observed in secondary lymphoid organs, excluding the spleen [8,9]. HEVs are characterized by the expression of peripheral node addressin and high levels of 6-sulfo sialyl Lewis X ligands. They are identified by the HEV-specific antibody MECA-79, which mediates the adherence and transendothelial migration of lymphocytes along the HEV vessel wall [10,11]. Interestingly, newly formed HEVs can appear in solid tumors associated with the distinct histological features of increased tumor infiltrative lymphocytes (TILs) in peri-HEVs or tertiary lymphoid tissue [12]. Previous studies have noted that a high density of HEVs in breast cancer and melanoma is associated with a favorable prognosis, possibly due to increased TILs and their phenotypes [13,14]. However, there have not been any combined analyses of HEVs and TILs conducted for gastric cancer.

In this study, we investigated HEVs and CD8+, Foxp3+, and CD20+ TILs, along with their correlation with the clinicopathologic features of AGC, and we evaluated the prognostic role of a combined analysis of HEVs and CD8+, Foxp3+, and CD20+ TILs.

## 2. Materials and Methods

### 2.1. Patients

In total, we enrolled 157 AGC patients who had undergone a radical resection between 2001 and 2005 at Ujeongbu St. Mary’s Hospital of the Catholic University of Korea. Patient data, including age, sex, diagnosis date, recurrence, and death, were retrieved from their electronic medical records. The pathologic features were re-evaluated by two board-certified pathologists (S.A.H. and H.W.H.), and TNM staging was reclassified according to the American Joint Committee on Cancer, 8th edition. This study was approved by the Institutional Research Ethics Board of Uijeongbu St. Mary’s Hospital of the Catholic University of Korea (UC20SESI0101) and was conducted in accordance with the tenets of the Declaration of Helsinki.

### 2.2. Immunohistochemistry

Tissue microarrays (TMAs) were constructed for immunohistochemistry. The tissue cores (2 mm) from two representative areas were punched and placed into the recipient blocks using a manual TMA device (SuperBioChips Laboratories, Seoul, Korea). Immunohistochemistry was performed using a Ventana Benchmark Autostainer (Ventana Medical System, Tucson, AZ, USA). Briefly, the slides were de-paraffinized, and antigen retrieval was conducted using heat-induced (92 °C for 30 min) epitope retrieval with an MC1 solution (Ventana Medical System, Tucson, AZ, USA). Sections were incubated with primary antibodies: CD8 (1:100, C8/144B, Dako, Cambridge, UK), Foxp3 (1:100, 236A/E7, Abcam, Cambridge, UK), CD20 (1:600, L26, Dako, Cambridge, UK), and MECA-79 (1:200, Santa Cruz, Tucson, AZ, USA). The Ultraview Polymer Detection Kit (Ventana Medical System, Tucson, AZ, USA) was used for visualization, and the stained sections were counterstained with hematoxylin. Human tonsil tissue was used as the positive control tissue. A negative control was performed by replacing the primary antibody with normal serum.

### 2.3. Evaluation of Immunohistochemistry

CD8+, Foxp3+, and CD20+ TILs were counted in five foci of the intratumoral area at a magnification of ×400 (BX51, Olympus, Tokyo, Japan). The mean numbers of CD8+, Foxp3+, and CD20+ TILs were manually calculated. MECA-79 expression was evaluated on the endothelial cells of vessels in the intratumoral stroma. The cut value for high- and low-expression was defined as the median value of CD8+, Foxp3+, and CD20+ TILs. The HEV level was denoted as positive if the expression of MECA-79 was found in any vessel.

### 2.4. Statistical Analysis

Categorical variables were compared, as needed, using the chi-square test and Fisher’s exact test. Overall survival (OS) was defined as the time from diagnosis to death of any cause or last follow-up. The survival time was plotted using the Kaplan–Meier method with a log rank test. Cox proportional hazards regression models were used to identify the significance of the prognostic factor. A two-sided *p* value of <0.05 was considered statistically significant in all tests and models. All data were analyzed using R statistical programming version 3.4.1 (http://www.r-project.org).

## 3. Results

### 3.1. Patient Characteristics

All patients underwent radical resection and lymph node dissection (D2). R0 resections were performed in 137 subjects (87.3%); 115 (71.4%) patients were men. The mean age was 66.3 years (range: 35–92 years). According to Lauren’s classification, 75 (47.8%) of the cancers were intestinal, 51 (32.5%) were diffuse, and 31 (19.7%) were mixed-type. At the time of resection, nodal and distant metastases were found in 99 (63.1%) and 17 (10.8%) patients, respectively. Based on the AJCC Cancer Staging Manual, 8th edition, 29 (18.5%) patients were classified as stage IB, 21 (13.4%) as stage IIA, 22 (14.0%) were classified as stage IIB, 17 (10.8%) were classified as IIIA, 16 (10.2%) were classified as stage IIIB, 35 (22.3%) were classified as stage IIIC, and 17 (10.8%) were classified as stage IV. The median OS was 43 months.

### 3.2. Correlation of HEVs and TILs with Clinicopathological Factors and Survival

The clinicopathologic parameters associated with HEVs and TILs are listed in Table 1. The median number of CD8+, Foxp3+, and CD20+ TILs was 46 (range: 2–412), 16 (range: 1–153), and 7 (range: 0–92), respectively. Based on the cut-value defined as the median number of TILs, a high CD8+ TIL level was found in 78 (49.7%) cases, a high Foxp3+ TIL level was found in 76 (48.4%) cases, and a high CD20+ TIL level was found in 69 (44.8%) cases (Figure 1A–F). HEVs, as evidenced by MECA-79 immunostaining, were observed in 60 (38.2%) cases (Figure 1G,H). A high CD8+ TIL level was significantly correlated with a low T stage (*p* = 0.045) and a low N stage (*p* = 0.011), while Foxp3+ TILs were only associated with a patient’s age (*p* = 0.035) (Table 1). A high CD20+ TIL level was not significantly related to any clinicopathologic parameter. HEVs were statistically correlated with perineural invasion (*p* = 0.005) (Table 1).

We analyzed the number of CD8+, Foxp3+, and CD20+ TILs and the presence of HEVs to clarify the association between the two. The number of CD8+ TILs was significantly higher in the HEV-positive group than in the HEV-negative group (*p* = 0.027) (Figure 2A). However, Foxp3+ and CD20+ TIL levels were similar in the HEV-positive and -negative groups (Figure 2B,C).

Survival analyses were conducted using the Kaplan–Meier method with the log rank test. Patients with a high CD8+ TIL level showed a significantly longer OS (*p* = 0.021) (Figure 3A). However, Foxp3+ TILs, CD20+ TILs, and HEVs were not significantly related with OS (Figure 3B–D).

### 3.3. Combined Analysis with CD8+ TIL and HEV.

A high CD8+ TIL level with HEVs was noted in 35 (22.3%) cases and was only associated with the low *n* stage (*p* = 0.048) (Table 2). OS was significantly different among the four groups based on a combined analysis of CD8+ TILs and HEVs (*p* = 0.015) (Figure 4A). The group with the high CD8+ and HEV levels showed the most favorable OS. In the high CD8+ TIL subgroup, the group with HEVs showed a significantly longer OS than that without HEVs (*p* = 0.016) (Figure 4B). However, HEVs were not significantly related to OS in the low CD8+ subgroup (*p* = 0.518) (Figure 4C).

### 3.4. Univariate and Multivariate Analysis

Following a univariate Cox’s proportional hazards analysis, a shorter OS was found to be significantly associated with an older age (<66 years vs. ≥66 years), a larger tumor size (<5.5 cm vs. ≥5.5 cm), a high T stage (pT2–3 vs. pT4), nodal metastasis (pN0 vs. ≥pN1), distant metastasis (pM0 vs. pM1), lymphovascular invasion, and resection margin (R0 vs. >R1). A longer OS was significantly associated with a high CD8+ TIL level (hazard ratio (HR), 0.611; confidence interval (CI), 0.400–0.933; *p* = 0.023) and a high CD8+ TIL level with HEVs (HR, 0.399; CI, 0.217–0.735; *p* = 0.003). After multivariate analysis, a high CD8+ TIL level with HEVs was found to be an independent favorable prognostic factor in AGC (*p* = 0.011, HR = 0.4352; 95% CI = 0.2453–0.8372) (Table 3).

## 4. Discussion

In this study, a combined analysis of HEVs, demonstrated by MECA-79 expression, and CD8+ TILs was associated with favorable clinicopathologic factors. Notably, a high CD8+ TIL level with HEVs was an independent prognostic factor for AGC.

HEVs have a crucial role in attracting TILs. The finding was derived from the association of HEVs and TILs, especially T-cells in human solid cancers, including breast cancer, melanoma, diffuse sclerosing variant of papillary thyroid carcinoma, and testicular seminoma [13,14,15,16]. Malignant melanoma and invasive ductal carcinoma with HEVs have demonstrated favorable prognoses [13,14]. However, no study exploring whether the level of TIL according to presence/absence of HEVs has a survival effect in GC has been performed.

Analyzing immune cell infiltration is the classic methodology for understanding the tumor immune microenvironment. The core role of TILs is a tumoricidal effect of CD8+ TILs [17]. Furthermore, from simple numerical estimation of TILs, studies examining how the tumor microenvironment controls TILs or boosts immunity have been recently conducted [18,19]. However, the interactions between the vascular network and TILs in AGC have not been extensively studied.

HEVs are postcapillary venules in lymphoid tissues and support lymphocyte migration from the blood [9]. The development and prognostic role of HEVs could differ according to the primary site of the tumor [14,16,20]. Based on our data, HEVs themselves did not have a significant prognostic role in AGC.

The evolution of HEVs partly matches the development of the tertiary lymphoid tissue. Though tertiary lymphoid tissue has survival benefits for breast, colorectal, and lung cancers, several studies have failed to determine their role in the survival associated with some solid tumors [20,21]. Our results, per previous research, indicated that the presence of tertiary lymphoid tissue without an analysis of immune cell components cannot be used to determine the prognosis of AGC [22].

The results of our combined analysis of HEVs and CD8+ TILs supported the important interaction between CD8+ TILs and HEVs in tumor immunology. The high density of CD8+ TIL in the intra-tumoral area was associated with high T and *n* stages. After univariate analysis, a high CD8+ TIL level was found to be a favorable prognostic factor (HR, 0.611; CI, 0.400–0.933; *p* = 0.023). However, a multivariate analysis did not demonstrate the value of CD8+ TILs as an independent prognostic factor. A high CD8+ TIL level with HEVs was identified as an independent and good prognostic factor (HR, 0.4532; CI, 0.2453–0.8372; *p* = 0.011494). Even in the high CD8+ TIL level group, HEVs could be used to segregate patients by survival (*p* = 0.016). There are two hypotheses for these results. The first is that HEVs themselves have the potential to attract and transport CD8+ T cells into a tumor. This was supported by the significant difference in the number of CD8+ TILs between the HEV-positive and -negative groups in our study. This is also consistent with previous research that has indicated that a high density of HEVs is associated with an upregulation of the genes associated with Th1 and cytotoxic T cells but not of genes associated with Th2 and Th17 [14]. In addition, common inciting factors, such as tumor necrosis factor, were found for both increasing CD8+ T cell and HEV levels [8,23].

The second hypothesis is that HEVs select highly competent CD8+ TILs with antitumor immunity. HEVs are well known to attract and transport naïve and central memory T cells into tumor tissues [24]. Previous reports have emphasized that effector cells developed from naïve T cells at the tumor site have higher potential and better long-term, effective antitumor responses than central memory T cells [25,26]. Naïve T cells priming in the vicinity of the tumor have an opportunity to transform into tumor antigen-specific and -sensitive T cells by a high antigenic load and a vast repertoire of tumor antigens [27,28]. Further research is required to determine the competence of CD8+ TILs associated with HEVs in GC. However, we cautiously suggest that the prognostic effect in AGC is related to HEVs with highly competent CD8+ TILs.

Foxp3 is a transcriptional factor that regulates the function and development of regulatory T cells (Treg) [29]. Research on a variety of solid cancers has reported that a high number of Foxp3+ TILs is associated with a poor prognosis due to the Treg-mediated suppression of antitumor immunity [30,31,32]. However, there are conflicting data regarding the role of Foxp3 TILs in the prognosis of gastric cancer [33,34]. Perrone et al. reported that an increased density of intratumoral foxp3+ TILs is associated with shorter survival time in stages II–IV gastric cancer patients [33]. However, several studies have failed to demonstrate a prognostic role of Foxp3+ TILs [34]. A single immunostaining for Foxp3 cannot identify effector or true Tregs because Foxp3+ TILs represent subpopulations of effector Tregs, naïve Tregs, and non-Tregs [35,36]. Our findings that Foxp3 TILs have no impact on the OS of AGC are consistent with previous research.

Notably, HEVs were not impacted by Foxp3 TIL changes in our cohort. Our results revealed that Foxp3+ TIL density in melanoma was similar for low- and high-density HEVs [13]. In addition, both CD8+ and Foxp3+ TILs were outnumbered by CD20+ TILs in GC with HEVs in our study. This result was consistent with a previous study in that CD3+ T cells comprised the main population in the perivascular space of HEVs when compared with CD20+ B cells. Furthermore, the presence of HEVs was also not correlated with the increased number of CD20+ TILs [37]. Based on our results, the function of HEVs as a gateway for TILs may be selective depending on the type of immune cell, particularly favoring CD8+ TILs but not Foxp3+ and CD20+ TILs.

The limitations of this retrospective study include the lack of the validation of different immune cells associated with HEVs and the colocalization between HEVs and TILs in the tumor microenvironment. Thus, further studies are required in this regard, and using flowcytometry and immunofluorescence could be useful to recognize different TIL populations associated with HEVs and to identify the spatial association between HEVs and TILs.

## 5. Conclusions

This study highlighted the prognostic role of HEVs and TILs in AGC. HEVs were found to be significantly associated with a high CD8+ TIL level, but they were not affected by Foxp3+ and CD20+ TILs in GC. However, HEVs themselves did not impact survival. A subgroup analysis demonstrated that a high CD8+ TIL level with HEVs demonstrated a favorable prognosis and was an independent prognostic factor in AGC. Further research on the interaction between CD8+ TILs and HEVs is needed to better understand the tumor microenvironment in AGC.

## Figures and Tables

**Figure 1 jcm-09-02628-f001:**
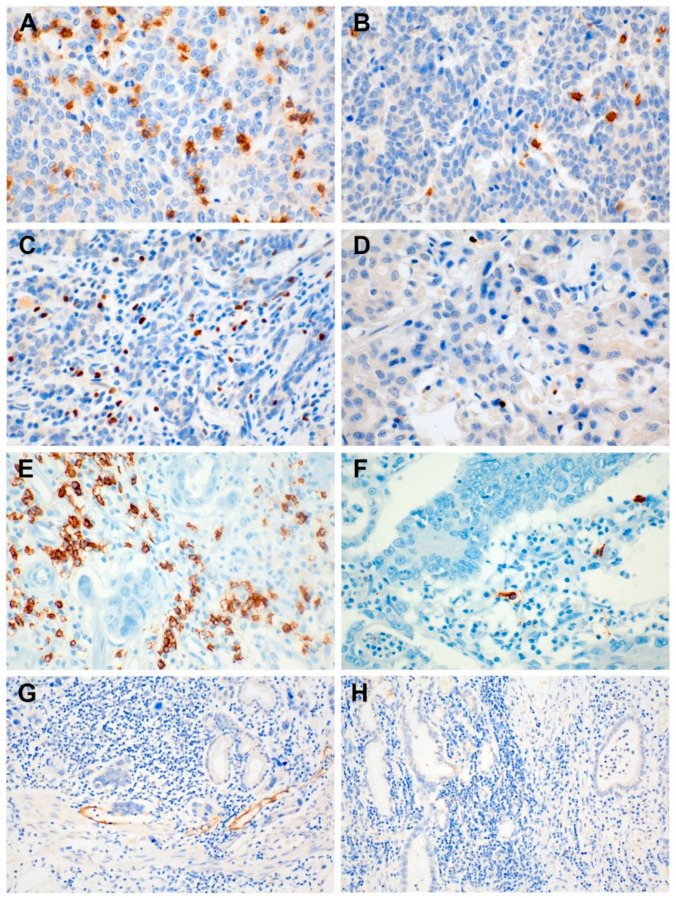
Representative immunohistochemistry of CD8+ tumor infiltrating lymphocytes (TILs), Foxp3+ TILs, and MECA-79 expression. (**A**) A high CD8+ TIL level, (**B**) a low CD8+ TIL level, (**C**) a high Foxp3+ TIL level, (**D**) a low Foxp3+ TIL level, (**E**) a high CD20+ TIL level, (**F**) a low CD20+ TIL level, and (**G**) a high endothelial venule (HEV) level exhibited by MECA-79; (**H**) no HEVs identified by MECA-79.

**Figure 2 jcm-09-02628-f002:**
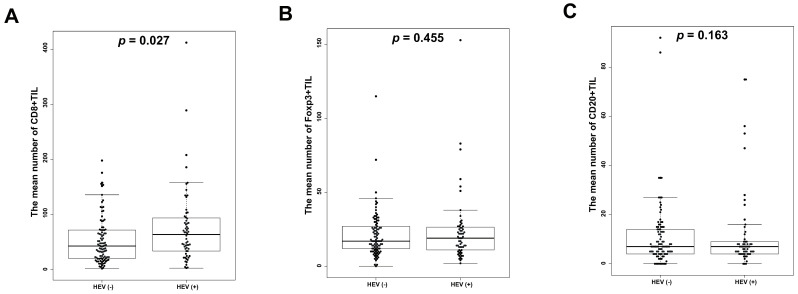
Differences in the mean number of TILs according to the presence or absence of HEVs in gastric cancer. (**A**) A significant elevation of the mean number of CD8+ TILs in HEV-positive patients compared to HEV-negative patients (*p* = 0.027); Non-significant increases of the mean number of Foxp3+ TILs (**B**) (*p* = 0.455) and CD20+ TILs (**C**) (*p* = 0.163) in HEV-positive cases as compared to HEV-negative cases.

**Figure 3 jcm-09-02628-f003:**
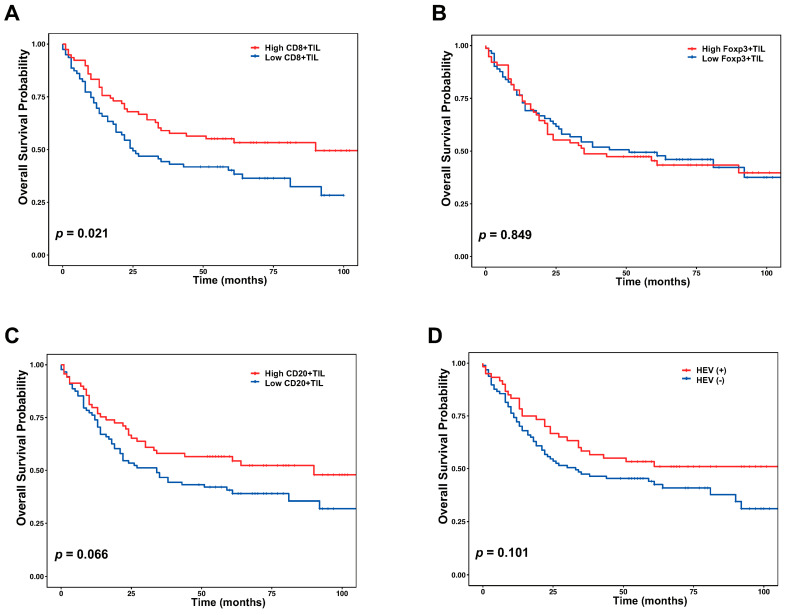
Prognostic analysis for CD8+ TILs, Foxp3+ TILs, CD20+ TILs, and HEVs in gastric cancer. (**A**) The Kaplan–Meier curve demonstrates a longer overall survival (OS) in patients with a high CD8+ TIL density. The significant differences of OS in patients were not observed according to the level of (**B**) Foxp3+ TILs, (**C**) CD20+ TILs, and (**D**) the presence or absence of HEVs.

**Figure 4 jcm-09-02628-f004:**
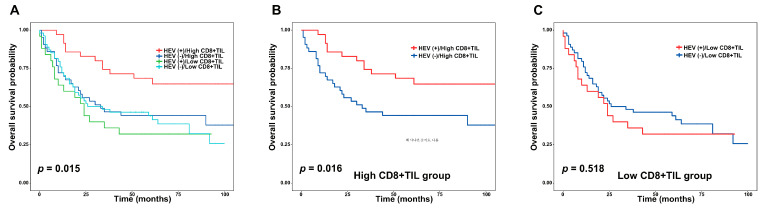
Prognostic effects for combined analysis of CD8+ TILs and HEVs. (**A**) Patients with a high CD8+ TIL density and HEVs demonstrated a longer OS than patients with a non-high CD8+TIL density and HEVs (*p* = 0.015). (**B**) In patients with a high CD8+ TIL density, those with HEVs showed a favorable OS (*p* = 0.016). (**C**) An OS difference was not observed between patients with/without HEVs in the low CD8+ TIL level group (*p* = 0.518).

**Table 1 jcm-09-02628-t001:** The correlation of clinicopathologic findings of advanced gastric cancer with tumor-infiltrating lymphocytes and HEVs.

	CD8			FOXP3			CD20			HEV		
Variables	High (*n* = 78) (%)	Low (*n* = 79) (%)	*p* Value	High (*n* = 74) (%)	Low (*n* = 83) (%)	*p* Value	High (*n* = 69) (%)	Low (*n* = 88) (%)	*p* Value	Positive (*n* = 60) (%)	Negative (*n* = 97) (%)	*p* Value
Age			0.923			0.035 *			0.594			0.749
≥66	47 (60.3)	46 (58.2)		52 (68.4)	41 (50.6)		43 (62.3)	50 (56.8)		37 (61.7)	56 (57.7)	
<66	31 (39.7)	33 (41.8)		24 (31.6)	40 (49.4)		26 (37.7)	38 (43.2)		23 (38.3)	41 (42.3)	
Gender			0.342			0.509			0.728			0.591
Male	54 (69.2)	61(77.2)		58 (76.3)	57 (70.4)		52 (75.4)	63 (71.6)		42 (70.0)	73 (75.3)	
Female	24 (30.8)	18 (22.8)		18 (23.7)	24 (29.6)		17 (24.6)	25 (28.4)		18 (30.0)	24 (24.7)	
Tumor size			0.092			0.918			0.994			0.116
≥5.5	31 (39.7)	43 (54.4)		35 (46.1)	39 (48.1)		32 (46.4)	42 (47.7)		23 (38.3)	51 (52.6)	
<5.5	47 (60.3)	36 (45.6)		41 (53.9)	42 (51.9)		37 (53.4)	46 (52.3)		37 (61.7)	46 (47.4)	
Histologic grade			0.169			0.266			0.838			0.083
Well	11 (14.1)	5 (6.3)		10 (13.2)	6 (7.4)		7 (10.1)	9 (10.2)		8 (13.3)	8 (8.2)	
Moderate	28 (35.9)	25 (31.6)		28 (36.8)	25 (30.9)		25 (36.2)	28 (31.8)		14 (23.3)	39 (40.2)	
Poorly	39 (50.0)	49 (62.0)		38 (50.0)	50 (61.7)		37 (53.6)	51 (58.0)		38 (63.3)	50 (51.5)	
Lauren classification			0.096			0.146			0.132			0.454
Intestinal	42 (53.8)	33 (41.8)		41 (53.9)	34 (42.0)		35 (50.7)	40 (45.5)		25 (41.7)	50 (51.5)	
Diffuse	19 (24.4)	32 (40.5)		19 (25.0)	32 (39.5)		17 (24.6)	34 (38.6)		21 (35.0)	30 (30.9)	
Mixed	17 (21.8)	14 (17.7)		16 (21.1)	15 (18.5)		17 (24.6)	14 (15.9)		14 (23.3)	17 (17.5)	
pT stage			0.045 *			0.910			0.181			1.000
T2	49 (62.8)	36 (45.6)		42 (55.3)	43 (53.1)		42 (60.9)	43 (48.9)		32 (53.3)	53 (54.6)	
T3/T4	29 (37.2)	43(54.4)		34 (44.7)	38 (46.9)		27 (30.1)	45 (51.1)		28 (46.7)	44 (45.4)	
pN stage			0.011 *			0.849			0.503			0.427
N0	37 (47.4)	21 (26.6)		27 (35.5)	31 (38.3)		28 (40.6)	30 (34.1)		25 (41.7)	33 (34.0)	
N1/N2/N3	41 (52.6)	58 (73.4)		49 (64.5)	50 (61.7)		41 (59.4)	58 (65.9)		35 (58.3)	64 (66.0)	
pM stage			0.318			0.708			1.000			0.291
M0	72 (92.3)	68 (86.1)		69 (90.8)	71 (87.7)		62 (89.9)	78 (88.6)		56 (93.3)	84 (86.6)	
M1	6 (7.7)	11 (13.9)		7 (9.2)	10 (12.3)		7 (10.1)	10 (11.4)		4 (6.7)	13 (13.4)	
Lymphovascular invasion			0.217			0.830			0.202			0.962
Yes	47 (60.3)	56 (70.9)		51 (67.1)	52 (64.2)		41 (59.4)	62 (70.5)		40 (66.7)	63 (64.9)	
No	31 (39.7)	23 (29.1)		25 (32.9)	29 (35.8)		28 (40.6)	26 (29.5)		20 (33.3)	34 (35.1)	
Perineural invasion			0.174			0.170			0.058			0.005 *
Yes	33 (42.3)	43 (54.5)		32 (42.1)	44 (54.3)		42 (60.9)	39 (44.3)		38 (63.3)	38 (39.2)	
No	45 (57.7)	36 (45.6)		44 (57.9)	37 (45.7)		27 (30.1)	48 (55.7)		22 (36.7)	59 (60.8)	

*, Significant at the level of *p* < 0.05.

**Table 2 jcm-09-02628-t002:** Association between a high CD8 level with a positive HEV level and a clinicopathologic finding of advanced gastric cancer.

	High CD8 with Positive HEV		
Variables	Yes (*n* = 35) (%)	No (*n* = 122) (%)	*p* Value
Age			0.765
≥66	22 (62.9)	71 (58.2)	
<66	13 (37.1)	51 (41.8)	
Gender			0.174
Male	22 (62.9)	93 (76.2)	
Female	13 (37.1)	29 (23.8)	
Tumor size			0.125
≥5.5	12 (34.4)	62 (50.8)	
<5.5	23 (65.7)	60 (49.2)	
Histologic grade			0.146
Well	4 (11.4)	12 (9.8)	
Moderate	7 (20.0)	46 (37.7)	
Poorly	24 (68.6)	64 (52.5)	
Lauren classification			0.137
Intestinal	15 (42.9)	60 (49.2)	
Diffuse	9 (25.7)	42 (34.4)	
Mixed	11 (31.4)	20 (16.4)	
T stage			0.832
T2/T3	20 (57.1)	65 (53.3)	
T4	15 (42.9)	57 (46.7)	
*n* stage			0.048 *
N0	19 (54.3)	39 (32.0)	
≥N1	16 (45.7)	83 (68.0)	
Lymphovascular invasion			1.000
Yes	23 (65.7)	80 (65.6)	
No	12 (34.3)	42 (29.1)	
Perineural invasion			0.831
Yes	18 (51.4)	58 (47.5)	
No	17 (48.6)	64 (52.5)	

*, Significant at the level of *p* < 0.05.

**Table 3 jcm-09-02628-t003:** Univariate and multivariate analyses of factors associated with overall survival of advanced gastric cancer.

Clinicopathologic Factors	Univariate Analysis			Multivariate Analysis		
	HR	95% CI	*p* Value	HR	95% CI	*p* Value
Age (years), ≥66 vs. <66	1.720	1.098–2.696	0.018 *	1.967	1.249–3.096	0.003 *
Gender, female vs. male	0.843	0.533–1.333	0.465			
Tumor size, ≥5.5 vs. <5.5	1.837	1.202–2.808	0.005 *	1.059	0.668–1.678	0.808
Histologic grade, poorly vs. well/moderately	1.286	0.593–2.788	0.524			
pT stage, pT4 vs. pT2/3	2.726	1.768–4.204	<0.001 *	1.880	1.165–3.033	0.010 *
pN stage, ≥pN1 vs. pN0	4.371	2.531–7.549	<0.001 *	2.972	1.668–5.295	<0.001 *
pM stage M1 vs. M0	4.079	2.408–6.910	<0.001 *	1.894	1.077–3.330	0.027 *
Lymphovascular invasion, Yes vs. No	3.229	1.898–5.494	<0.001 *	1.504	0.792–2.857	0.212
Perineural invasion, Yes vs. No	1.457	0.957–2.217	0.079			
R0 resection No vs. Yes	2.973	1.729–5.109	<0.001 *	2.120	0.663–6.780	0.205
CD8 High vs. Low	0.611	0.400–0.933	0.023 *	0.799	0.507–1.254	0.327
FOXP3 High vs. Low	0.887	0.576–1.335	0.539			
CD20 High vs. Low	0.669	0.431–1.030	0.068			
HEV Yes vs. No	0.693	0.444–1.081	0.106			
High CD8/ positive HEV Yes vs. No	0.399	0.217–0.735	0.003 *	0.464	0.243–0.883	0.019 *
High FOXP3/ Positive HEV Yes vs. No	0.661	0.379–1.153	0.145			
High CD20/ positive HEV Yes vs. No	0.540	0.271–1.078	0.081			

HR, hazard ratio; CI, confidence interval; HEV, high endothelial venule; *, Significant at the level of *p* < 0.05.

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
