# Peer review of "High Endothelial Venule with Concomitant High CD8+ Tumor-Infiltrating Lymphocytes Is Associated with a Favorable Prognosis in Resected Gastric Cancer"

_jcm, 2020, doi:10.3390/jcm9082628_

Round 1
Reviewer 1 Report
In the manuscript titled "High endothelial venule with concomitant high CD8+ tumor-infiltrating lymphocytes is 3 associated with a favorable prognosis in resected gastric cancer," have determined the prognostic significance of CD8+ and Foxp3+TILs in combination with high endothelial venules (HEV) in advanced gastric cancer patients. The prognostic significance of HEV has been earlier studied in breast, colon, oral and melanoma cancer but authors have reported it in gastric cancer for the first time. This study can become interesting, if authors will consider following comments.
- In methods section describe in detail immunohistochemistry protocol. They should included counting/scoring criteria in a separate heading.
- It will be good, if authors can do immunofluorescence staining to presence of HEV and immune cells.
- It will be good, if authors can correlate expression of HEV with other immune cells.
- All the graphical results should be improved as axis labelling and figure legends are not visible.
- In discussion, section they should include studies from other cancers too.
Author Response
Dear Reviewers and Editors,
We thank you very much for your consideration of our manuscript, entitled “High endothelial venule with concomitant high CD8+ tumor-infiltrating lymphocytes is associated with a favorable prognosis in resected gastric cancer” (Manuscript ID: jcm-862250), and for the opportunity to resubmit a revised version. We also appreciate the reviewers’ helpful comments on our manuscript. After discussions among all the co-authors, we have substantially revised the manuscript to reflect the reviewers’ concerns. Changes in the text are highlighted in yellow. Point-by-point responses to specific reviewer comments are indicated below. The approved number (UC20SESI0101) of the Institutional Research Ethic Board are also added.
Reviewer 1
- 1. In methods section describe in detail immunohistochemistry protocol. They should included counting/scoring criteria in a separate heading.
As recommended by the reviewer, we have added detailed immunohistochemistry protocol in method section. The contents for counting TIL and scoring criteria for TILs are described in a separate heading. Corresponding contes are as follows:
Line 84-105:
2.2. Immunohistochemistry
Tissue microarrays (TMA) were constructed for immunohistochemistry. The tissue cores (2 mm) from two representative areas were punched and placed into the recipient blocks using a manual TMA device (SuperBioChips Laboratories, Seoul, Korea). Immunohistochemistry was performed using Ventana Benchmark Autostainer (Ventana Medical System, Tucson, AZ, USA). Briefly, the slides were de-paraffinized, and antigen retrieval was conducted using heat-induced (92°C for 30 min) epitope retrieval with MC1 solution (Ventana Medical System, Tucson, AZ, USA). Sections were incubated with primary antibodies: CD8 (1:100, C8/144B, Dako, Cambridge, UK), Foxp3 (1:100, 236A/E7, Abcam, Cambridge, UK), CD20 (1:600, L26, Dako, Cambridge, UK), and MECA-79 (1:200, Santa Cruz, Tucson, AZ, USA). The Ultra view polymer detection kit (Ventana Medical System, Tucson, AZ, USA) was used for visualization, and the stained sections were counterstained with hematoxylin. Human tonsil tissue was used as positive control tissue. Negative control was performed by replacing the primary antibody with normal serum.
2.3 Evaluation of immunohistochemistry
CD8+, Foxp3+, and CD20+ TILs were counted in five foci of the intratumoral area at a magnification of ×400 (BX51, Olympus, Tokyo, Japan). The mean numbers of CD8+, Foxp3+, and CD20+ TILs were calculated manually. MECA-79 expression was evaluated on the endothelial cells of vessels in the intratumoral stroma. The cut value for high- and low-expression was defined as the median value of CD8+, Foxp3+, and CD20+ TILs. HEV was positive if the expression of MECA-79 was found in any vessel.
- It will be good, if authors can do immunofluorescence staining to presence of HEV and immune cells.
We agree the reviewer’s opinion that immunofluorescence can be helpful method for our study. In our study, we aimed to analyze the HEV and immune cells by practical method that is commonly used in routine pathological practice. Regarding to this issue, immunohistochemistry is the most suitable method that is well established and easily applicable to routine practice. However, although this thing, immunofluorescence is one of valuable methods to validate our results. Thus, we describe the limitation of our research in which immunofluorescence has not been done. Corresponding contest are as follows:
Line 305-310
The limitations of this retrospective study include lacking validation of different immune cells associated with HEV and colocalization between HEV and TILs at once in the tumor microenvironment. Thus, further studies are required in this regard, and using flowcytometry and immunofluorescence could be useful to recognize different TIL populations associated with HEV and to identify the spatial association between HEV and TILs.
- It will be good, if authors can correlate expression of HEV with other immune cells.
As recommended by the reviewer, we have evaluated CD20+ tumor-infiltrating lymphocytes and the analyzed the association between HEV and CD20+TIL. However, we cannot find a prognostic role of CD20+ TIL and association between CD20+TIL and HEV. We added the corresponding content in Figures 1 to 3, tables 1 and 3, abstract, methods, results and discussion section.
Line 28
Abstract
HEV was identified in 60 (38.2%) cases and was significantly associated with an increased number of CD8+ TIL (P = 0.027) but not of Foxp3+ TIL (P = 0.455) and CD20+ TIL (P = 0.163).
Line 88 to 100
Material and Methods
Sections were incubated with primary antibodies: CD8 (1:100, C8/144B, Dako, Cambridge, UK), Foxp3 (1:100, 236A/E7, Abcam, Cambridge, UK), CD20 (1:600, L26, Dako, Cambridge, UK), and MECA-79 (1:200, Santa Cruz, Tucson, AZ, USA). The Ultra view polymer detection kit (Ventana Medical System, Tucson, AZ, USA) was used for visualization, and the stained sections were counterstained with hematoxylin. Human tonsil tissue was used as positive control tissue. Negative control was performed by replacing the primary antibody with normal serum.
2.3 Evaluation of immunohistochemistry
CD8+, Foxp3+, and CD20+ TILs were counted in five foci of the intratumoral area at a magnification of ×400 (BX51, Olympus, Tokyo, Japan). The mean numbers of CD8+, Foxp3+, and CD20+ TILs were calculated manually.
Line 129 to 147
The median number of CD8+, Foxp3+ and CD20+ TILs was 46 (range, 2–412), 16 (range, 1–153), and 7 (range, 0–92), respectively. Based on the cut-value defined as the median number of TIL, high CD8+ TIL was found in 78 (49.7%) cases, high Foxp3+ TIL in 76 (48.4%) cases, and high CD20+ TIL in 69 (44.8%) cases (Figures 1A to 1F). HEVs as evidenced by MECA-79 immunostaining were observed in 60 (38.2%) cases (Figures 1G to 1H). High CD8+ TIL was significantly correlated with a low T stage (P = 0.045) and N stage (P = 0.011), while Foxp3+ TIL was only associated with a patient’s age (P = 0.035) (Table 1). High CD20+ TIL was not significantly related to any clinicopathologic parameter. HEV was statistically correlated with perineural invasion (P = 0.005) (Table 1).
We analyzed the number of CD8+, Foxp3+, and CD20+ TILs and the presence of HEV to clarify the association between the two. The number of CD8+ TIL was significantly higher in the HEV-positive group than in the HEV-negative group (P = 0.027) (Figure 2A). However, Foxp3+ and CD20+ TIL levels were similar in HEV-positive and negative groups (Figures 2B and 2C).
Survival analyses were conducted using the Kaplan–Meier method with the log rank test. Patients with high CD8+ TIL showed significantly longer OS (P = 0.021) (Figure 3A). However, Foxp3+ (P = 0.849), CD20+ TILs, and HEV were not significantly related with OS (Figures 3B to 3D).
Line 296 to 304
Notably, HEV was not impacted by Foxp3- TIL changes in our cohort. Our results reveal that Foxp3+ TIL density in melanoma is similar for low- and high-density HEV [13]. In addition, both CD8+ and Foxp3+ TILs were outnumbered than CD20+ TIL in GC with HEV in our study. This result was consistent with a previous study in that CD3+ T cells comprised the main population in the perivascular space of HEV when compared with CD20+ B cells. Furthermore, the presence of HEV was also not correlated with the increased number of CD20+ TIL [38]. Based on our results, the function of HEV as the gateway for TILs can be selective for the type of immune cell, particularly favoring CD8+ TIL but not Foxp3+ and CD20+ TILs.
- All the graphical results should be improved as axis labelling and figure legends are not visible.
We improved the graphical results through making axis labelling clearly and rechecked figure legends.
- In discussion, section they should include studies from other cancers too.
As recommended by the reviewer, we added the studies from other cancers in discussion. Corresponding content is followed as:
Line 233-239
HEV has a crucial role in attracting TIL. The finding is derived from the association of HEV and TIL, especially T-cell in human solid cancer, including breast cancer, melanoma, diffuse sclerosing variant of papillary thyroid carcinoma, and testicular seminoma [13–16]. Malignant melanoma and invasive ductal carcinoma with HEV demonstrated favorable prognosis [13, 14]. However, no study exploring whether the level of TIL according to presence/absence of HEV has a survival effect in GC has been performed.

Reviewer 2 Report
The present study investigated on the role of high endothelial venules (HEVs), CD8+ tumor-infiltrating lymphocytes (TILs), and Foxp3+TILs and their correlation with clinicopathologic features of advanced gastric cancer and the prognostic role of a combined analysis of HEV, CD8, and Foxp3+TILs.
In order to be published the authors need to address some concerns:
1) Regarding the mean number of CD8+ and Foxp3 TILs I would mention the possibility to analyze by flow cytometry the percentage of different T-cell populations then correlate to the HEVs high.
2)The literature could be updated.
3)The paper should be checked by an English mother tongue
Author Response
Dear Reviewers and Editors,
We thank you very much for your consideration of our manuscript, entitled “High endothelial venule with concomitant high CD8+ tumor-infiltrating lymphocytes is associated with a favorable prognosis in resected gastric cancer” (Manuscript ID: jcm-862250), and for the opportunity to resubmit a revised version. We also appreciate the reviewers’ helpful comments on our manuscript. After discussions among all the co-authors, we have substantially revised the manuscript to reflect the reviewers’ concerns. Changes in the text are highlighted in yellow. Point-by-point responses to specific reviewer comments are indicated below. The approved number (UC20SESI0101) of the Institutional Research Ethic Board are also added.
Reviewer 2
- Regarding the mean number of CD8+ and Foxp3 TILs I would mention the possibility to analyze by flow cytometry the percentage of different T-cell populations then correlate to the HEVs high.
We agree that flow cytometry can be useful to examine and validate different immune cell populations in our study. However, we used archive formalin-fixed paraffin embedded blocks that were limited materials for flow cytometry. Thus, we describe the limitation of our study that flow cytometry cannot be performed to examine TILs in discussion section.
Line 305 to 310
The limitations of this retrospective study include lacking validation of different immune cells associated with HEV and colocalization between HEV and TILs at once in the tumor microenvironment. Thus, further studies are required in this regard, and using flowcytometry and immunofluorescence could be useful to recognize different TIL populations associated with HEV and to identify the spatial association between HEV and TILs.
- The literature could be updated.
As recommended by the reviewer, we updated the references through the revised manuscript
- The paper should be checked by an English mother tongue
As recommended by the reviewer, the manuscript has been revised and rechecked by a native speaker.

Round 2
Reviewer 2 Report
I reviewed the work in the new version, the authors worked on the points required during the first revision and I can accept the paper with these changes
Author Response
We thank you very much for your consideration of our manuscript, entitled “High endothelial venule with concomitant high CD8+ tumor-infiltrating lymphocytes is associated with a favorable prognosis in resected gastric cancer” (Manuscript ID: jcm-862250)